# Impact of the 294 bp SINE Insertion in 5′UTR of the *GLYATL3* Gene on Gene Expression and Phenotypic Variation

**DOI:** 10.3390/ani15101375

**Published:** 2025-05-09

**Authors:** Chenyu Zhou, Suwei Qiao, Yao Zheng, Miao Yu, Hong Chen, Cai Chen, Ali Shoaib Moawad, Bo Gao, Chengyi Song, Xiaoyan Wang

**Affiliations:** 1College of Animal Science and Technology, Yangzhou University, Yangzhou 225009, China; mx120230888@stu.yzu.edu.cn (C.Z.); mx120230885@stu.yzu.edu.cn (S.Q.); mz120180996@yzu.edu.cn (Y.Z.); mz120221567@stu.yzu.edu.cn (M.Y.); mx120230878@stu.yzu.edu.cn (H.C.); 007302@yzu.edu.cn (C.C.); ali.shoeib@agr.kfs.edu.eg (A.S.M.); bgao@yzu.edu.cn (B.G.); cysong@yzu.edu.cn (C.S.); 2Department of Animal Production, Faculty of Agriculture, Kafrelsheikh University, Kafrelsheikh 33516, Egypt; 3International Joint Research Laboratory in Universities of Jiangsu Province of China for Domestic Animal Germplasm Resources and Genetic Improvement, Yangzhou 225009, China

**Keywords:** pig, *GLYATL3*, SINE, enhancer, growth trait

## Abstract

This study explored how a specific DNA fragment (SINE) inserted near the *GLYATL3* gene affects gene expression and physical traits in pigs. Researchers tested 15 pig breeds and found that the SINE insertion was absent in hybrid pigs but present in all purebreds. Pigs with the SINE insertion showed differences in growth, particularly when reaching 30 kg in weight. Experiments revealed that the SINE insertion boosts *GLYATL3* gene activity in the brain and enhances the function of a gene promoter linked to development. These findings suggest that this SINE insertion may influence traits like growth by regulating *GLYATL3* expression in pigs.

## 1. Introduction

The glycine N-acyltransferase-like 3 (*GLYATL3*) gene encodes a protein that belongs to the N-acyltransferase family [1,2,3]. The protein may be involved in the metabolic processes of amino acids and fatty acids in biochemical pathways and has demonstrated *GLYATL3* as an effective epigenetic marker of primary breast cancer [4,5]. Members of the N-acyltransferase family typically function by catalyzing the conjugation of amino acids with fatty acids or other molecules [6], forming acyl amino acids or other conjugated products and playing a role in physiological functions such as fat metabolism, amino acid metabolism, drug metabolism, energy metabolism, fat storage and release, hormone regulation, cellular protection, and microbial interactions [7]. Up to now, there was no report about the function of this gene in the economic traits of domestic animals.

Retrotransposons, including short interspersed nuclear elements (SINEs), long interspersed nuclear elements (LINEs), and long terminal repeat (LTR) sequences, are important components of mammalian transposable elements (TEs) [8,9,10,11,12,13,14]. They are widely distributed in mammalian genomes and comprise approximately 37% of the pig genome [15]. Because of their copy-paste mechanism during mobilization, retrotransposons generate plentiful polymorphism throughout the genomes. Unable to remove themselves from their insertion sites, this characteristic of unidirectional integration provides significant benefits for reconstructing pedigrees and phylogenies, as the ancestral state is much clearer than with most other types of genetic polymorphisms. Their potential mobilization has a significant impact on the structure and function of these genomes [16]. Providing cis-regulatory elements, retrotransposons can activate or inactivate proximal host genes and even cause phenotypic variations in domestic animals [17,18,19,20,21]. Retrotransposon insertion polymorphisms (RIPs) in the pig genome can be used to assess genetic variations and population structure among different pig breeds [22,23].

SINEs are one type of the most frequently found DNA repetitive sequences in the eukaryotic genome [8,9]. They rely on enzymes encoded by other elements, such as LINEs, for transcription and insertion [24]. Compared with LINEs and LTRs, SINE is shorter in length and the most widely distributed element in mammalian genomes, potentially having the most significant genomic and genetic impact [9,15,25,26]. Previous studies have shown that SINE-RIP can affect the phenotypes of mammals such as merle phenotypes in dogs [27] and the back-fat phenotype in pigs [28,29]. SINEs, ranging from 150 to 300 bp, are thought to be more compatible with their host organisms than other retrotransposon types, such as LTRs and LINEs. They have the ability to co-evolve with host genomes and can significantly influence gene structure and genome evolution [30]. Therefore, SINE-RIP may be important tools for studying biodiversity and genetics, and even for molecular breeding in domestic animals.

Using the Mobile Element Locator Tool (MELT, v2.2.2) [31], a bioinformatics tool, numerous RIPs were identified [32], one of which is SINE-RIP located in the 5′UTR region of the pig *GLYATL3*. Given the significant role of *GLYATL3* and the extensive applications of RIPs, the polymerase chain reaction (PCR) was used in this study to determine the polymorphism of SINE insertions in the 5′UTR region of *GLYATL3* across different pig breeds. Moreover, we attempted to determine whether there was an association between phenotype and genotype.

## 2. Materials and Methods

### 2.1. Animals and Samples

A total of 796 individuals from 15 pig breeds (3 commercial, 2 hybrid, and 10 native) were used to examine SINE-RIP. Detailed information including breeds, type, etc., can be found in Appendix A.

### 2.2. DNA and RNA Extraction

Porcine genomic DNA was isolated from ear tissue samples using the TIANamp Genomics DNA Extraction kit (TIANGEN, Beijing, China). For transcriptomic analysis, total RNA was extracted from cerebellum, pallium, liver, and back-fat tissues with the FastPure Complex Cell/Tissue Total RNA Isolation kit (Vazyme, Nanjing, China), in accordance with standard manufacturer protocols. Nucleic acid quantification was performed using an Implen GmbH NanoPhotometer N60 Touch spectrophotometer (Munich, Germany) to determine DNA and RNA concentrations. Long-term preservation of biomolecules was achieved through cryogenic storage, with DNA aliquots maintained at −20 °C and RNA samples preserved at −80 °C.

### 2.3. Verification of RIPs and Genotyping Using PCR

Primers were designed based on the flanking regions of the SINE insertion site in the genome using Oligo 7.0 software (Appendix A).

The PCR amplification was conducted in 20 µL reaction volumes containing 50 ng genomic DNA template and 10 pmol of each primer pair. The reaction mixture was prepared with 10 µL 2× TaqMix (Vazyme, Nanjing, China) supplemented with nuclease-free water to achieve final concentrations. Thermal cycling parameters comprised an initial denaturation at 95 °C (5 min), followed by 30 cycles of denaturation (95 °C, 30 s), primer-specific annealing (30–60 s), and extension (72 °C, 30 s), concluding with a terminal elongation step at 72 °C (5 min). Post-amplification products were temporarily preserved at 4 °C prior to analysis. Amplicon validation was performed by 1.5% agarose gel electrophoresis using ethidium bromide staining, with subsequent digital documentation through automated gel imaging systems.

### 2.4. Expression Analysis Using qPCR

Genotyping of 23 Mi pigs was conducted, and RNA from 4 tissues (cerebellum, pallium, liver, and back fat) of 9 Mi pigs was selected, with 3 individuals per genotype. cDNA was reverse-transcribed using a FastKing RT kit (with gDNase) (TIANGEN, Beijing, China). Subsequently, qPCR was performed using a 7900 HT Fast Real-Time PCR System (Applied Biosystems, New York, NY, USA). The total reaction mixture included 10 µL of SYBR mix (Vazyme, Nanjing, China), 0.4 µL each of the forward and reverse primer, 1 µL of cDNA sample, and 8.2 µL of ddH2O. Gene expression was normalized to GADPH, and the results were analyzed using the 2^−∆∆Ct^ method.

### 2.5. Vector Construction

To evaluate the activity of the SINE insertion in *GLYATL3*, the total 294 bp insertion of SINE was cloned. Next, Promoter2.0 (https://services.healthtech.dtu.dk/services/Promoter-2.0/, accessed on 11 March 2024) was used to predict the likely promoter locations of *GLYATL3*. We predicted a promoter region with a score > 1 located from 2017 bp upstream of the SINE insertion. Therefore, the SINE + 2017 bp fragment with a total length of 2311 bp was also cloned.

The 2 fragments were separately inserted into the pGL3-basic vector, resulting in the construction of the PGL3-SINE− and PGL3-SINE+ vectors.

To functionally validate the enhancer potential of the SINE insertion, Oct4 and Myc minimal promoter elements were subcloned from their respective pTol2-mCherry donor vectors (Oct4: pTol2-Oct4-mCherry; Myc: pTol2-Myc-mCherry) [33], respectively, and inserted into the PGL3-basic vector. The new vectors were renamed as Oct4-SINE− and Myc-SINE−. Subsequently, the 294 bp SINE insertion was introduced into both the vectors, resulting in the construction of Oct4-SINE+ and Myc-SINE+. The primers used for vector construction are shown in Appendix A.

### 2.6. Dual-Luciferase Reporter Assay

CHO-K1 and HeLa cells were cultured, and a total of 2 × 10^5^ cells were transferred to a 6-well plate for transfection. Plasmids were separately transfected into the cultured CHO-K1 and HeLa cells, and after 48 h, luciferase activity was determined using the Dual-Luciferase Reporter Assay System (Vazyme, Nanjing, China). A Modulus™ II multimode microplate reader was used for measurements (Turner Biosystems, Sunnyvale, CA, USA) following the manufacturer’s instructions. Each experiment was performed independently three times.

CHO-K1 and HeLa cells were cultured in Dulbecco’s Modified Eagle Medium containing 10% fetal bovine serum, 100 U/mL penicillin, and 0.1 mg/mL streptomycin. All cells were cultured in a humidified environment at 37 °C in an atmosphere of 5% carbon dioxide. All cell culture reagents were procured from Thermo Fisher Scientific (Waltham, MA, USA).

### 2.7. Statistical Analysis

Growth parameters encompassing body weight, back-fat thickness, and age at reaching a weight of 30 kg and 100 kg were systematically evaluated in Largewhites (*n* = 458) selected based on predictive growth performance markers. Live weights at slaughter were recorded via electronic weighing, paired with age documentation at standardized 30 kg and 100 kg thresholds. Ultrasonographic assessment of subcutaneous adipose tissue was performed using the LX8000B system (Beijing Kang Cheng Da Technology Co., Ltd., Beijing, China) at two physiological endpoints—100 kg live weight and post-slaughter—with measurements taken at the third–fourth intercostal space.

Popgene32 was used to evaluate genotype frequencies, allele frequencies, and Hardy–Weinberg equilibrium. The formula for calculating polymorphic information content (PIC) [34] is as follows:$$\mathrm{PIC}=1-\sum_{i=1}^{m}Pi^{2}-\sum_{i=1}^{m-1}\sum_{j=i+1}^{m}2Pi^{2}Pj^{2}$$

The experimental data were statistically analyzed using the SPSS 17.0 software package (SPSS Inc., Chicago, IL, USA). Comparisons were performed by one-way analysis of variance (ANOVA) followed by Tukey’s post hoc test, with data expressed as the mean ± standard deviation (SD). To address multiple comparisons, error rates were controlled through Bonferroni correction, establishing a stringent significance threshold at *p* < 0.05.

## 3. Results

### 3.1. RIP Generated by SINE Insertion in Pig GLYATL3

Many RIPs in pig genome have been previously obtained using MELT. They were further annotated using RepeatMasker (v4.1.8), and one 294 bp SINE-RIP was predicted in the 5′UTR region of pig *GLYATL3* located at 43,641,443–43,659,754 bp in chromosome 7 of the porcine reference genome (Sscrofa11.1). This SINE insertion could be classified under the SINEA1 subfamily. The insertion polymorphism was identified using PCR with the DNA pool as a template (Figure 1A) and based on the electrophoretogram. The insertion of SINE has been widely found in Duroc, Landrace, Largewhite, Huai, Mi, Jiangquhai, Wuzhishan, and Bama, and it has formed a relatively high polymorphism. This RIP was localized in 668 bp downstream of the transcription start site (TSS) and 1908 bp upstream of the gene start codon (ATG) (Figure 1B). Three genotypes were determined in the electrophoretogram as homozygous for insertion, heterozygous for insertion, and homozygous without insertion, which corresponded to SINE+/+, SINE+/−, and SINE−/−, respectively. This insertion polymorphism was designated as *GLYATL3*-SINE-RIP.

### 3.2. Distribution and Genetic Diversity of GLYATL3-SINE-RIP in Different Pig Breeds

RIPs were detected in 15 different pig populations, including Duroc, Landrace, Largewhite, Sushan, Sujiang, Erhualian, Jiangquhai, Mi, Meishan, Min, Rongchang, Tibetan, Jinhua, Wuzhishan, and Bama. In addition, the Hardy–Weinberg test was conducted and PIC was calculated to jointly evaluate the genetic diversity of the RIP. SINE insertion was found in all pig breeds (Table 1). Specifically, SINE+/+ was common, having a genotype frequency of >0.45 and a relatively high frequency in Jiangquhai, Meishan, and Largewhite. Interestingly, SINE+/+ was not found in Sujiang and Sushan. Mi, Wuzhishan, and Bama were analyzed and they diverged from the Hardy–Weinberg equilibrium. Moderate polymorphism was exhibited in most pig breeds except for Sujiang, Sushan, Erhualian, and Rongchang.

### 3.3. Correlation Between GLYATL3-SINE-RIP and the Phenotype of Largewhite

PCR was carried out using DNA from 458 Largewhite individuals, and genotypes were decided to analyze the relationship between genotypes and the phenotypic data including body weight before slaughter, back-fat thickness, age (days) at 30 kg body weight, age (days) at 100 kg body weight, and back-fat thickness at 100 kg body weight. As shown in Table 2, significant differences (*p* < 0.05) between SINE+/+ individuals and SINE−/− individuals were found in age (days) needed to reach 30 kg body weight. These findings demonstrated that SINE insertion may decrease the age (days) of Largewhite pigs needed to reach 30 kg body weight.

### 3.4. Effect of GLYATL3-SINE-RIP Insertion on GLYATL3 Expression in Mi Pigs

To further investigate the effect of SINE insertion on *GLYATL3* in pigs, qPCR was used to analyze differences in gene expression among different genotypes (SINE+/+, SINE+/−, and SINE−/−) in Mi pigs. qPCR revealed significant differences (*p* < 0.05) in the cerebellar tissue of Mi pigs (Figure 2); specifically, a significant increase in *GLYATL3* expression was observed in individuals with SINE+/+ compared with those with SINE−/−. In contrast, significant differences were not found between these groups for other tissues, including the back fat, pallium, and liver.

### 3.5. Validation of the Enhancer of SINE Insertion in the Pig GLYATL3 Gene

To explore the potential effect of SINE insertion in the *GLYATL3* gene, a dual-luciferase reporter assay was used to determine its effect on regulatory activity. A promoter region with a score > 1 was predicted (Figure 3A) using the online tool Promoter2.0, which was located 2017 bp upstream of the 294 bp SINE insertion, which was approximately 1500 bp upstream of the TSS. As shown in Figure 3A, the predicted 2017 bp region, potentially containing the promoter, along with or without the 294 bp SINE insertion, were inserted into the pGL3 vector to validate the promoter and were named pGL3-SINE+ and pGL3-SINE−, respectively. As shown in Figure 3D, an increasing trend was observed in the pGL3-SINE+ and pGL3-SINE− vectors. Therefore, to further verify the regulatory activity of SINE, two mini promoters (Oct4 and Myc) were inserted into the pGL3 vector and named Oct4-SINE− and Myc-SINE−, respectively. The 294 bp SINE was then inserted into each of the reconstructed vectors and named Oct4-SINE+ and Myc-SINE+, respectively (Figure 3C). A significant difference (*p* < 0.05) was observed between Oct4-SINE+ and Oct4-SINE− (Figure 3E). Specifically, Oct4-SINE+ showed significantly enhanced activity compared with Oct4-SINE−. Overall, *GLYATL3*-SINE-RIP insertion may exhibit enhancer activity for the mini promoter Oct4.

## 4. Discussion

*GLYATL3* is a member of the N-acyltransferase family [1,6]. The proteins of this family likely play a role in the metabolic pathways of amino acids and fatty acids [2,3,4,5]. *GLYATL3*-SINE-RIP was one of the RIPs found in our previous study [32] that was located in the 5′UTR region of *GLYATL3*. This SINE insertion belongs to the SINEA1 subfamily, which is considered to be one of the youngest types [15,35]. Previous studies have confirmed that SINEA1 has a greater tendency to generate polymorphisms in both intragenic and extragenic regions of the pig genome [15]. Furthermore, SINEA1 found in CA5B [29] and LEPROT [36] had a significant effect on gene expression. Therefore, as a young retrotransposon, SINEA1 is more likely to have an effect on gene expression and also affect phenotype variations in domestic animals when the insertion generates polymorphisms.

The distribution and genetic diversity of *GLYATL3*-SINE-RIP among 15 pig breeds including commercial, hybrid, and native breeds were identified using PCR [22]. SINE insertion was detected to be predominant in commercial pig breeds and some native pig breeds, whereas a few SINE insertions were detected in hybrid pig breeds and some native pigs. Hybrid pig breeds may experience higher selection pressure and the insertion may be discarded during the selection process. Moreover, 80% of the populations were in Hardy–Weinberg equilibrium, and 20% of the pig breeds (Mi, Wuzhishan, and Bama) deviated from the Hardy–Weinberg equilibrium. This deviation could be attributed to the limited population sizes and nonrandom mating patterns in some local pig breeds.

Furthermore, phenotypic data of 458 Largewhite pigs were used for association analysis with SINE insertion genotypes. SINE+/+ individuals developed faster (*p* < 0.05), and their weight of 30 kg was achieved faster compared with SINE−/− individuals. Although studies on *GLYATL3* are ongoing, previous studies have shown that GLYAT is a member of the glycine transferase family and that its expression is significantly correlated with bone size and body thinness [37]. The reason for this significant phenomenon may be that the period before piglets reach their weight of 30 kg is a critical stage for the development of bone size and body thinness. During this period, bone development and body thinness can affect their growth rate [38,39,40,41]. However, the reason that there was no significant difference from SINE+/+ to SINE−/− individuals when they reached 100 kg is that their bone development slowed down. qPCR was used to determine *GLYATL3* expression in the different tissues of individuals with the SINE+/+, SINE+/−, and SINE−/− genotypes. Our results indicated that SINE insertion upregulated *GLYATL3* expression in the cerebellum, a tissue where its expression has been previously unreported in the NCBI database (https://www.ncbi.nlm.nih.gov/gene/100516087, accessed on 14 March 2024). Considering the crucial role of the cerebellum in neurodevelopment and motor coordination, SINE insertion may affect the growth and development processes of individuals by altering *GLYATL3* expression patterns. Further investigations into the regulatory mechanisms of SINE-mediated *GLYATL3* expression and its functional role in the cerebellum are therefore warranted.

Based on our study, the effects of *GLYATL3*-SINE-RIP insertion on the expression and phenotype suggest a regulatory role of this SINE insertion. And previous research has suggested that SINE may affect adjacent gene promoters [42]. Therefore, a dual-luciferase reporter gene assay was used to confirm the effect of SINE insertion on the activity of the promoter of *GLYATL3* and two mini promoters containing Oct4 and Myc [43,44]. A significant increase (*p* < 0.05) in activity was ultimately found for the mini promoter Oct4 in the CHO-K1 cell line. Another mini promoter, Myc, showed the opposite phenomenon compared with the Oct4 promoter, without any significant improvement trend. Moreover, the insertion of SINE even led to a decrease in its activity. This phenomenon was also observed in our previous study on PPARγ [28]. It is possible that the insertion fragment contains specific inhibitory regulatory factors that suppress the activity of the mini promoter Myc. Additionally, no significant enhancement trend was observed in the CHO-K1 cell line. This could be attributed to the length of the 2017 bp fragment that was amplified in our study, which might contain certain factors that collectively inhibit promoter activity and participate in regulatory mechanisms. The lack of a significant enhancement trend in the HeLa cell line can be explained on the basis of the high *GLYATL3* expression in the ovary, as seen in the NCBI expression profile, versus its relatively low expression in cervical cells. This indicates that SINE insertion has an enhancer-regulatory effect on the mini promoter Oct4. The role of TE as a regulatory component influencing gene expression on PPARγ has been previously confirmed [28]. Furthermore, the study by Zheng et al. [29] delves into the structural analysis of SINEs to explore the potential mechanisms via which these elements might function as enhancers, and SINE insertion has also been shown to act as an enhancer that regulates expression and influences traits [45]. Based on previous research and SINE insertion in the 5′UTR region found in this study, which impacts the expression pattern of *GLYATL3* and the growth rate of Largewhites, it could be inferred that SINE functions as an enhancer that regulates *GLYATL3* expression.

Actually, as one important quantitative trait, growth trait is not only affected by one loci of polymorphism. In our previous research, we identified numerous RIPs and we are developing a liquid-phase chip based on these RIPs mining the whole genome of pigs. Notably, the significant finding that *GLYATL3*-SINE-RIP functions as an enhancer not only highlights the potential role of RIPs in gene regulation but also provides valuable insights for the subsequent development of our liquid-phase chip. These discoveries will enhance their effective application in genomic research and breeding programs.

## 5. Conclusions

One SINE-RIP located in the 5′UTR region of *GLYATL3* was identified in this study based on comparative genomics and validation using PCR. *GLYATL3*-SINE-RIP exhibits differential distribution between hybrid and native pigs, with this RIP exhibiting a significant correlation with growth rate in Largewhites. The observed disparity in growth traits is likely attributed to the enhancer activity of SINE, which significantly enhances the expression of *GLYATL3*. The positive result of *GLYATL3*-SINE-RIP provides a reference for us in using genetic selection to optimize growth traits in porcine breeding programs.

## Figures and Tables

**Figure 1 animals-15-01375-f001:**
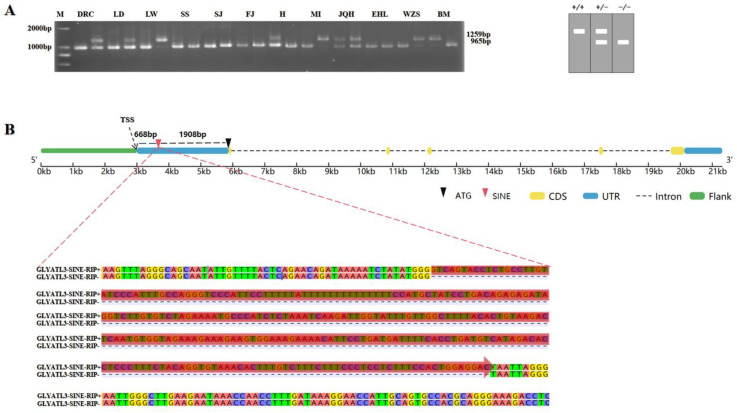
Electrophoretogram and structural diagram of SINE-insertion polymorphism in the 5′UTR region of pig *GLYATL3*. (**A**) Pooled DNA from 12 breeds of pigs was used for PCR detection. M, DNA Maker DL2000; DRC, Duroc; LD, Landrace; LW, Largewhite; SS, Sushan; SJ, Sujiang; FJ, Fengjing; H, Huai; MI, Mi; JQH, Jiangquhai; EHL, Erhualian; WZS, Wuzhishan; BM, Bama; (**B**) RIP location in *GLYATL3*.

**Figure 2 animals-15-01375-f002:**
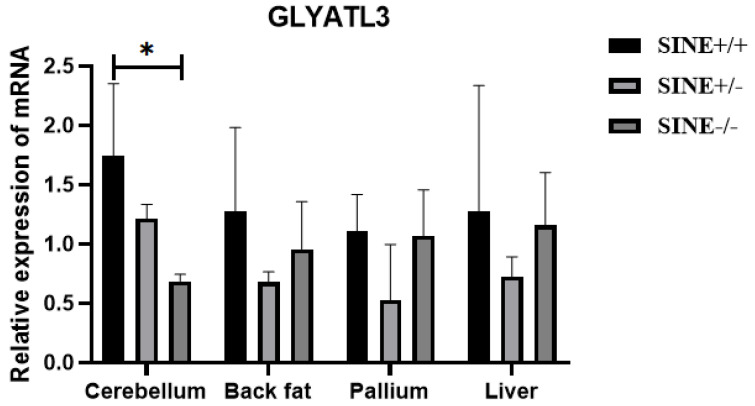
Relative expression of *GLYATL3* in the tissues of Mi pigs. * *p* < 0.05.

**Figure 3 animals-15-01375-f003:**
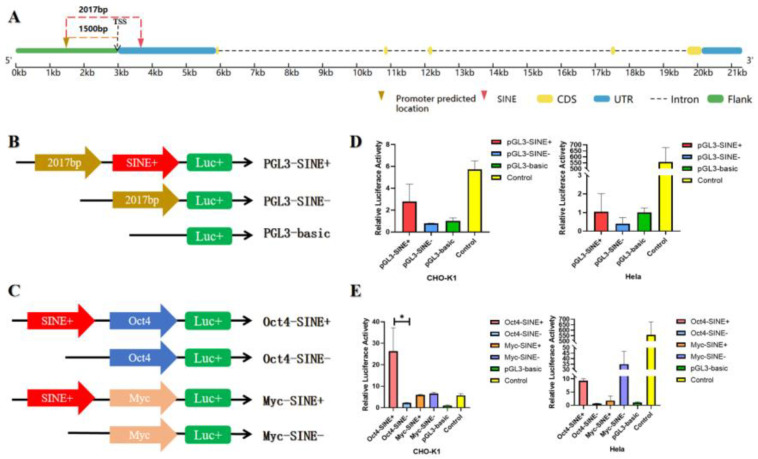
Impact of SINE on regulatory activity. (**A**) Predicted position of the promoter region in *GLYATL3*, and 294 bp SINE insertion located 2017 bp downstream. (**B**) Vector construction to evaluate the impact of *GLYATL3*-SINE-RIP insertion on the regulatory activity of 2017 bp containing its own promoter. (**C**) Vector construction to evaluate the impact of *GLYATL3*-SINE-RIP insertion on the regulatory activity of the mini promoter containing Oct4 and MYC. (**D**,**E**) Dual-luciferase reporter assay for each vector from (**B**,**C**) in CHO-K1 and Hela cell lines. * *p* < 0.05.

**Table 1 animals-15-01375-t001:** *GLYATL3*-SINE-RIP polymorphism detection among different pig breeds.

RIP	Breeds	Number	+/+	+/−	−/−	+	−	*p* Value	*PIC*
*GLYATL3*-SINE-RIP	DRC	24	0.25	0.58	0.17	0.54	0.46	0.39	0.37
	LD	23	0.17	0.61	0.22	0.49	0.51	0.29	0.38
	LW	458	0.48	0.41	0.11	0.67	0.33	0.47	0.34
	SS	24	0	0.08	0.92	0.04	0.96	0.83	0.08
	SJ	24	0	0.08	0.92	0.04	0.96	0.83	0.08
	EHL	24	0	0.17	0.83	0.08	0.92	0.66	0.14
	JQH	23	0.48	0.52	0	0.74	0.26	0.10	0.31
	MI	29	0.14	0.69	0.17	0.49	0.51	0.04	0.38
	MS	24	0.5	0.29	0.21	0.64	0.36	0.08	0.36
	MZ	24	0.17	0.29	0.54	0.31	0.69	0.16	0.34
	RC	24	0	0.33	0.67	0.17	0.83	0.33	0.24
	TB	24	0.13	0.29	0.58	0.27	0.73	0.20	0.32
	JH	24	0.29	0.38	0.33	0.48	0.52	0.22	0.38
	WZS	23	0.29	0.17	0.54	0.35	0.65	<0.01	0.35
	BM	24	0.33	0.25	0.42	0.46	0.54	0.02	0.37

DRC, Duroc; LD, Landrace; LW, Largewhite; SS, Sushan; SJ, Sujiang; EHL, Erhualian; JQH, Jiangquhai; MI, Mi; MS, Meishan; MZ, Min; RC, Rongchang; TB, Tibetan; JH, Jinhua; WZS, Wuzhishan; BM, Bama.

**Table 2 animals-15-01375-t002:** Genotypic and phenotypic association analysis of 458 Largewhites.

Genotype	Number	Body Weight Before Slaughter (kg)	Thickness of Back Fat (mm)	Age at 30 kg Body Weight (Days)	Age at 100 kg Body Weight (Days)	Back-Fat Thickness at 100 kg Body Weight (mm)
*GLYATL3*-SINE+/+	220	100.52 ± 9.11	10.79 ± 2.83	74.57 ± 7.63 a	162.37 ± 7.94	10.70 ± 2.55
*GLYATL3*-SINE+/−	190	100.04 ± 8.97	10.83 ± 2.94	74.74 ± 7.25 ab	163.19 ± 8.51	10.80 ± 2.71
*GLYATL3*-SINE−/−	48	99.07 ± 10.19	11.17 ± 3.31	77.39 ± 8.68 b	163.29 ± 7.95	11.20 ± 2.88

No significant difference exists between groups with the same letter, whereas a significant difference exists between groups with different letters (*p* < 0.05).

## Data Availability

All data needed to evaluate the conclusions in this paper are present either in the main text or the Appendix A.

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
