# Peer review of "Impact of the 294 bp SINE Insertion in 5′UTR of the GLYATL3 Gene on Gene Expression and Phenotypic Variation"

_animals, 2025, doi:10.3390/ani15101375_

Round 1

Reviewer 1 Report

Comments and Suggestions for Authors

The study investigates the effects of a SINE insertion in the 5’ UTR region of the GLYATL3 gene on gene expression and phenotypic traits in pigs. Before being considered for publication, improvements and clarifications need to be made.

Here are my comments and suggestions:

1) Abstract and Intro: Please consider including the implications of the findings for breeding programs and/or genetic selection.
2) Abstract: Please re-phrase "These findings suggest that this SINE insertion may influence traits like growth by regulating GLYATL3 expression in pigs".
3) Intro: Please include statements that support your study's novelty/significance. What is the gap? How this study addresses this gap?
4)  The transition between explaining SINEs as retrotransposons and their relevance to genetic variation in pigs needs to be improved.
5) M&M: Mention the number of animals per breed used for the analysis. Include details about the statistical power of the analyses if available.
6) M&M: The statistical analysis description is not detailed enough. Clarify which model was used.
7) M&M: How were the associations between genotypes and phenotypes tested? 
8) Were multiple testing corrections applied? What were the threshold criteria for significance (e.g., p-value cut-off)?
9) Results: Provide a descriptive statistical analysis of the data by breed.
10) Discussion: Discuss how the findings could be applied in practical breeding programs for pigs. Are there any potential limitations?
11) Discussion/Conclusion: Elaborate more on the implications of finding an association between SINE insertion and growth traits. For example, what would be the next step in practical breeding programs?
12) Please double-check formatting and citations. 
13) Please double-check gene names and units in the manuscript.
14) This manuscript has a high percentage of matched words in the iThenticate report. Most of those repeated words come from a paper from the same group. Please rephrase the methodology section.

Author Response

Response to Reviewer 1 Comments

  • Abstract and Intro: Please consider including the implications of the findings for breeding programs and/or genetic selection.

Response 1:Thank you for your suggestion. As a new type marker, this SINE insertion polymorphism may assist genetic selection to optimize growth traits in porcine breeding programs. We have added this part in the part of Abstract and Introduction.

  • Abstract: Please re-phrase "These findings suggest that this SINE insertion may influence traits like growth by regulating GLYATL3 expression in pigs".

Response 2:Thank you for your suggestion. We have rewritten the sentences.

  • Intro: Please include statements that support your study's novelty/significance. What is the gap? How this study addresses this gap?

Response 3:Thank you for your suggestion. Retrotransposons are unable to remove themselves from their insertion sites, and this characteristic of unidirectional integration provides significant benefits for reconstructing pedigrees and phylogenies, as the ancestral state is much clearer than with most other types of genetic polymorphisms. Compared with AFLP markers based on SNPs across various crop species,retrotransposon insertion polymorphisms are more informative(reviewed by R Kalendar). SINEs, which typically exist as short fragments ranging from 150-300 bp, are thought to be more compatible with their host organisms than other retrotransposon types, such as LTRs and LINEs. They have the ability to co-evolve with host genomes and can significantly influence gene structure and genome evolution(reviewed by Platt II). Therefore SINE-RIP may be important tools for studying biodiversity and genetics, and even for molecular breeding in domestic animals.The relevant content has been added to the introduction of the manuscript.

R Kalendar , A J Flavell , T H N Ellis , T Sjakste , C Moisy , A H= Schulman Analysis of plant diversity with retrotransposon-based molecular markers Heredity (Edinb). 2010 Aug 4;106(4):520–530. doi: 10.1038/hdy.2010.93

Roy N Platt II, Michael W Vandewege , David A Ray. Mammalian transposable elements and their impacts on genome evolution. Chromosome Res. 2018 Feb 1;26(1):25–43. doi: 10.1007/s10577-017-9570-z

  • The transition between explaining SINEs as retrotransposons and their relevance to genetic variation in pigs needs to be improved.

Response 4:Thank you for your suggestion. We have rewritten the sentences and improved our explanation.

  • M&M: Mention the number of animals per breed used for the analysis. Include details about the statistical power of the analyses if available.

Response 5:Thank you for your comment. All animals per breed used for the analysis were provided in Table S1. Details about the statistical power of the analyses have been added in 2.1 and 2.7.

  • M&M: The statistical analysis description is not detailed enough. Clarify which model was used.

Response 6:Thank you for your suggestion. The experimental data were statistically analyzed using SPSS 17.0 software package (SPSS Inc., Chicago, USA). Comparisons were performed by one-way analysis of variance (ANOVA) followed by Tukey's post hoc test, with data expressed as mean ± standard deviation (SD). We have added these details in the part of Statistical analysis.

  • M&M: How were the associations between genotypes and phenotypes tested?

Response 7:Thank you for your comment. We have added statistical analysis about the associations between genotypes and phenotypes in 2.7.

  • Were multiple testing corrections applied? What were the threshold criteria for significance (e.g., p-value cut-off)?

Response 8:Thank you for your remind. To address multiple comparisons, Error rates were controlled through Bonferroni correction, establishing a stringent significance threshold at p < 0.05.

  • Results: Provide a descriptive statistical analysis of the data by breed.

Response 9:Thank you for your suggestion. We have revised the relevant contents as required and provided a descriptive statistical analysis of the data by breed in 3.1 and 3.2.

  • Discussion: Discuss how the findings could be applied in practical breeding programs for pigs. Are there any potential limitations?

Response 10:Thank you for your comment. In our previous research, we identified numerous RIPs and aim to develop a liquid-phase chip utilizing these RIPs for RIP mining in whole genome of pigs. Notably, the significant finding that GLYATL3-SINE-RIP functions as an enhancer not only highlights the potential role of RIPs in gene regulation but also provides valuable insights for the subsequent development of our liquid-phase chip. These discoveries will enhance their effective application in genomic research and breeding programs.

  • Discussion/Conclusion: Elaborate more on the implications of finding an association between SINE insertion and growth traits. For example, what would be the next step in practical breeding programs?

Response 11:Thank you for your comment. GLYATL3-SINE-RIP exhibits differential distribution between hybrid and native pigs, with this RIP significant correlation with growth rate in Largewhites. The observed disparity in growth traits is likely attributed to the enhancer activity of SINE, which significantly enhances the expression of GLYATL3.The positive result of GLYATL3-SINE-RIP provides a reference for us to make liquid-phase chips for RIPs mining in the whole genome of pigs.

  • Please double-check formatting and citations.

Response 12:Thank you for your suggestion. We have checked formatting and citations.

  • Please double-check gene names and units in the manuscript.

Response 13:Thank you for your suggestion. We have checked gene names and units in the manuscript.

14) This manuscript has a high percentage of matched words in the iThenticate report. Most of those repeated words come from a paper from the same group. Please rephrase the methodology section.

Response 14:Thank you for your suggestion. We have rephrased the methodology section.

Reviewer 2 Report

Comments and Suggestions for Authors

This study focused on the effects of a SINE insertion in 5’ UTR of GLYATL3 gene on its expression and growth rate in pigs. The results indicated the potential of this insertion as a candidate molecular marker for growth rate in pig breeding. The manuscript basically meets the requirements of the journal, but there are still some problems that need to be addressed. The specific issues are as follows:

1. 15 pig breeds were selected to verify the genetic diversity. Why were these 15 specific breeds chosen?

2. Line118: “2−∆∆Ct method” should be replaced with the “2−∆∆Ct method”.

3. Line 158-159:a reference should be added following: “The formula for calculating polymorphic information content (PIC) is as follows:”

4. Figure 1: Genotypes need to be labeled in Figure 1.

5. All gene abbreviations mentioned in this article should be in italics.

6. The author reference format is not consistent, please standardize and revise your reference information systematically, reviewing each entry line by line.

7. The article did not mention the description of the test result of production traits of Large white, please add the description of the result.

8. Line 207-212: Similar explanation is wrong, I did not understand the author's meaning, this article does not involve the concept of genetic distance and Fst, please check it carefully.

9. The “5 UTR” in the text should be expressed as “5 'UTR”.

Author Response

Response to Reviewer 2 Comments

  1. 15 pig breeds were selected to verify the genetic diversity. Why were these 15 specific breeds chosen?

Response 1:Thank you for your comment.These 15 breeds include 3 commercial pigs, 2 hybrid pigs and 10 native pigs (including Taihu pigs and Jinhua pigs from the Yangtze River Basin, Min pigs from the north and miniature pigs from the south of China). The selection of these 15 kinds of pigs can fully reflect the genetic diversity changes of this locus.

  1. Line118: “2−∆∆Ct method” should be replaced with the “2−∆∆Ct method”.

Response 2:Thank you for you suggestion.We have modified the relevant content

  1. Line 158-159:a reference should be added following: “The formula for calculating polymorphic information content (PIC) is as follows:”

Response 3:Thank you for you suggestion.The relevant citation has been added

  1. Figure 1: Genotypes need to be labeled in Figure 1.

Response 4:Thank you for you suggestion.Genotypes have been labeled in Figure 1.

  1. All gene abbreviations mentioned in this article should be in italics.

Response 5:Thank you for you suggestion.We have checked formatting.

  1. The author reference format is not consistent, please standardize and revise your reference information systematically, reviewing each entry line by line.

Response 6:Thank you for you suggestion.We have checked formatting and citations.

  1. The article did not mention the description of the test result of production traits of Large white, please add the description of the result.

Response 7:Thank you for your comment.PCR was carried out using DNA from 458 Largewhite individuals, and genotypes were decided to analyze the relationship between genotypes and the phenotypic data including body weight before slaughter, back-fat thickness, age (days) at 30 kg body weight, age (days) at 100 kg body weight, and back-fat thickness at 100 kg body weight. As shown in Table 2, significant differences (P<0.05) between SINE+/+ individuals and SINE −/− individuals were found in age (days) to reach 30 kg body weight. These findings demonstrated that SINE insertion may decrease the age (days) of Largewhite pigs to reach 30 kg body weight..

  1. Line 207-212: Similar explanation is wrong, I did not understand the author's meaning, this article does not involve the concept of genetic distance and Fst, please check it carefully.

Response 8:We are sorry about our mistake.We have confirmed and modified the expression of this part.

  1. The “5 UTR” in the text should be expressed as “5 'UTR”.

Response 9:We are sorry about our mistake.We have modified the relevant content
